# Mental health and well-being of older adults living with HIV in sub-Saharan Africa: a systematic review

Patrick Nzivo Mwangala  ,[1,2] Adam Mabrouk,[1] Ryan Wagner,[3] Charles R J C Newton,[1,4] Amina A Abubakar[1,4,5,6]

¹Department of Clinical Research (Neurosciences), KEMRI-Wellcome Trust Research Programme, Kilifi, Kenya
²University of the Witwatersrand School of Public Health, Johannesburg, South Africa
³MRC/Wits Rural Public Health and Health Transitions Research Unit (Agincourt), School of Public Health, Faculty of Health Sciences, University of the Witwatersrand, Parkton, Gauteng, South Africa
⁴Department of Psychiatry, University of Oxford, Oxford, UK
⁵Institute for Human Development, Aga Khan University, Nairobi, Kenya
⁶Department of Public Health, Pwani University, Kilifi, Kenya

**Correspondence to**
Patrick Nzivo Mwangala;
pmwangala27@gmail.com

## ABSTRACT

**Objective** In this systematic review, we aimed to summarise the empirical evidence on common mental disorders (CMDs), cognitive impairment, frailty and health-related quality of life (HRQoL) among people living with HIV aged ≥50 years (PLWH50 +) residing in sub-Saharan Africa (SSA). Specifically, we document the prevalence and correlates of these outcomes.

**Design, data sources and eligibility criteria** The following online databases were systematically searched: PubMed, CINAHL, PsycINFO, Embase and Scopus up to January 2021. English-language publications on depression, anxiety, cognitive function, frailty and quality of life among PLWH50 + residing in SSA were included.

**Data extraction and synthesis** We extracted information, including study characteristics and main findings. These were tabulated, and a narrative synthesis approach was adopted, given the substantial heterogeneity among included studies.

**Results** A total of 50 studies from fifteen SSA countries met the inclusion criteria. About two-thirds of these studies emanated from Ethiopia, Uganda and South Africa. Studies regarding depression predominated (n=26), followed by cognitive impairment (n=13). Overall, PLWH50 + exhibited varying prevalence of depression (6%–59%), cognitive impairments (4%–61%) and frailty (3%–15%). The correlates of CMDs, cognitive impairment, frailty and HRQoL were rarely investigated, but those reported were sociodemographic variables, many of which were inconsistent.

**Conclusions** This review documented an increasing number of published studies on HIV and ageing from SSA. However, the current evidence on the mental and well-being outcomes in PLWH50 + is inadequate to characterise the public health dimension of these impairments in SSA, because of heterogeneous findings, few well-designed studies and substantial methodological limitations in many of the available studies. Future work should have sufficiently large samples of PLWH50+, engage appropriate comparison groups, harmonise the measurement of these outcomes using a standardised methodology to generate more robust prevalence estimates and confirm predictors.

**PROSPERO registration number** CRD42020145791.

## BACKGROUND

All societies globally are experiencing an ageing population—some are in its early stages, and some are more advanced.[1]

Likewise, the proportion of people living with HIV (PLWH) aged ≥50 years (considered as older adults within the HIV literature) is growing rapidly across the globe,[2] demanding an increase in research, policy and practice to address the complex needs of this vulnerable group.[3] In 2016, there were around 6 million PLWH50 + globally, representing 16% of the entire adult HIV population.[2] This proportion was projected to reach 21% by the end of 2020, with Southern and Eastern Africa containing the vast majority of PLWH50+.[2] The increase in older PLWH is mainly driven by two factors: effective antiretroviral therapy (ART) and new HIV infections among the elderly.[4]

This rapidly growing segment of PLWH50 + yields new challenges. Several HIV and ageing cohorts (mostly from high-income countries, HICs) are pointing out an elevated burden of physical conditions (eg, cardiometabolic diseases), mental morbidities (eg, depression, cognitive disorders), psychosocial challenges (eg, stigma, loneliness) and geriatric syndromes (eg, frailty)

among PLWH ≥50 years (PLWH50+) compared with their uninfected counterparts.[4–6] Unfortunately, the mechanisms for the heightened risk of poor outcomes among PLWH50 + are not fully understood but span a host of disease-related factors, legacy of the early years of the HIV epidemic and host vulnerability factors.[4] Frailty, common mental disorders (CMDs) such as depression and anxiety, and cognitive impairments are of particular concern in this population.[4–6] Frailty is an important emerging indicator of vulnerability that is increasingly being evaluated among individuals ageing with HIV.[7] It describes a state of physiological vulnerability that, in the presence of internal or external stressors, puts a person at an elevated risk of adverse clinical outcomes, such as hospitalisation, functional disability and mortality.[8 9] Overall, mental morbidities and frailty are understudied in the literature despite their elevated burden and potential adverse impacts on HIV treatment and health-related quality of life (HRQoL).

Despite the apparent need to sharpen the focus on the needs of the growing cohort of PLWH50+, this population is being neglected in HIV care and prevention efforts, especially in low-income and middle-income settings of sub-Saharan Africa (SSA). In SSA, this concern is not just based on the close association between ageing and physical, cognitive, mental impairments, but also on the recognition that older adults are not considered to be a priority for policy issues. They are marginalised and increasingly vulnerable to greater poverty.[10 11] Thus, the burden and determinants of physical and mental impairments in these adults in SSA, the region most affected by HIV globally, are largely undocumented.[12]

The course and implication of physical and mental morbidities among PLWH50 + are well described in HICs, but it is an emergent research area in SSA. However, since 2010, the SSA region has witnessed a rise in interest in HIV and ageing outcomes, evidenced by the establishment of cohort studies of older people in different parts of Africa.[13–15] This is encouraging since data from HICs may not be readily generalised to SSA where differences in, for example, genetics, social environmental milieu, formal and informal support systems will almost certainly alter the risk profile and morbidity characteristics among PLWH50+. Therefore, the current review aims to summarise the existing empirical evidence on CMDs, cognitive impairment, frailty and HRQoL among PLWH50 + in SSA by addressing the following specific objectives.
1. Establish the prevalence of CMDs, cognitive impairment and frailty among PLWH50 + in SSA.
2. Identify the correlates of CMDs, cognitive impairment, frailty and HRQoL among PLWH50 + in SSA.

## METHODS
We followed the Preferred Reporting Items for Systematic Reviews and Meta-analyses guidelines to report the study.[16]

### Search strategy
Five online databases (PubMed, CINAHL, PsycINFO, Embase and Scopus) were searched for publications. The last search was conducted on 4 January 2021. The following search terms were used: (HIV OR HIV-1 OR HIV/AIDS OR HIV infections) AND (adult OR older adult OR older people OR older individual OR elderly) AND (Africa OR sub-Saharan Africa OR Africa South of the Sahara) AND (cognitive impairment OR neurocognitive impairment OR neurological complication OR HIV-associated neurocognitive disorder OR common mental disorder OR depression OR depressive symptoms OR depressive disorder OR anxiety OR anxiety disorder OR quality of life OR health-related quality of life OR grip strength OR hand strength OR frailty). We also searched through the reference lists of included articles to access additional articles. Details of the full search strategy are provided in online supplemental file 1.

Studies were considered eligible if they: (1) involved PLWH50 + or had participants whose mean/median age was ≥50 years or aggregated their outcomes of interest by age (including the ≥50 age category); (2) were conducted within SSA; (3) were empirical and published in a peer-reviewed journal and (4) reported any of our outcomes of interest (CMDs, cognitive impairment, frailty or HRQoL). We excluded studies that did not aggregate the outcomes of interest by HIV status. We also excluded studies that were published in non-English languages. Two of the authors (PNM and AM) independently screened the titles, abstracts and full articles for eligibility and reached consensus.

### Data extraction and quality assessment
The two authors (PNM and AM) independently extracted the following study characteristics using a similarly designed extraction sheet: (1) first author; (2) country and the year of publication; (3) sample description; (4) how the outcome of interest was measured; (5) results reported; (6) key findings on the outcome of interest. For studies exclusively carried out among PLWH50+, we computed or extracted the reported percentages of the relevant outcomes. Similarly, for studies with middle-aged participants but whose mean/median age was ≥50 years, we also computed or extracted the reported percentages for the relevant outcomes for the whole sample. For the studies with middle-aged participants (but whose mean/median age was not at least 50 years), we computed percentages for participants who fell within the ≥50 category, which was usually provided in the papers. In some studies, it was impossible to calculate these percentages. Hence, the occurrence of a specific outcome was reported in the original effect measure, for example, OR, median or mean. Any disagreement between the reviewers during data abstraction was also resolved by consensus.

We evaluated the quality of included studies using the modified version of the Newcastle-Ottawa Scale.[17] The scale uses five parameters (sample size, sample representativeness, comparability between respondents and

non-respondents, ascertainment of the outcome of interest, and quality of descriptive statistics reporting) to obtain a total score ranging from 0 to 5. A score ≥3 is considered a low risk of bias, while a score of <3 is regarded as a high risk of bias.

## Data synthesis and analysis

A narrative synthesis approach was used to compare studies and describe the state of evidence on all reported outcomes; by describing the effect estimates, for instance, proportions, means with SD, and median with their IQRs. Where applicable, ranges were used to summarise the outcomes of interest reported. A meta-analysis could not be done due to a limited number of studies and substantial heterogeneity among studies.

## Patient and public involvement statement

The accompanying manuscript is part of broader doctoral research work for the first author. During the conceptual phase of this work, community engagement through meetings with different stakeholders was carried out to contextualise the issues of ageing and HIV at the coast of Kenya (eg, in terms of priorities and preferences). These stakeholders included the Kilifi and Mombasa counties' departments of Health, subcounty health management teams, and hospital management teams of various health facilities at the study setting. The research team received and incorporated useful feedback from these meetings towards the design and implementation of the project. It also led to further elaborations regarding ongoing community engagement during and after the study, including disseminating any work arising from this project using these forums and others such as publications, hospital departmental meetings, patient groups and conferences as an essential way to discuss the implications of the findings. For this systematic review, patients were not involved in the conduct of the study. However, the results will be disseminated to the relevant patient groups within the HIV clinics through feedback meetings.

## RESULTS

### Characteristics of included studies

We identified 8741 articles from the 5 online databases and an additional 7 citations from snowballing. Of these, 2693 were duplicates. Hence, we screened 6055 titles and abstracts for initial eligibility, out of which 575 articles were identified. Full articles were obtained for these citations, of which 50 met the eligibility criteria (figure 1). The eligible studies were conducted between 2006 and 2021 among 15 SSA countries of Uganda (n=13), South Africa (n=12), Ethiopia (n=8), Tanzania (n=5), Nigeria (n=3), Cameroon (n=2), Côte d'Ivoire (n=2), Senegal (n=2), Kenya (n=2), Botswana (n=1), Zambia (n=1), Malawi (n=1), Namibia (n=1), Togo (n=1) and Burkina Faso (n=1). However, some of these studies were conducted in multiple countries. Figure 2 shows the geographical distribution of these studies across the SSA region. Virtually

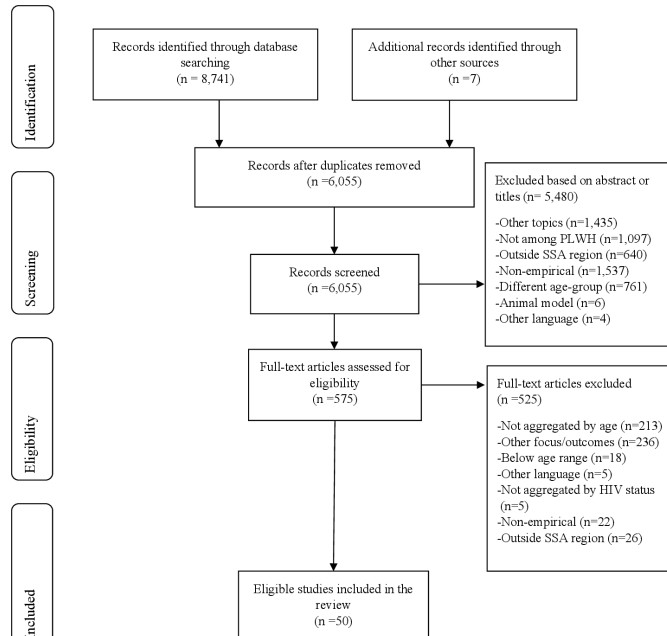

**Figure 1** Identification of eligible studies. PLWH, people living with HIV; SSA, sub-Saharan Africa.

all the included studies were cross-sectional, except one that presented data from a randomised trial.[18] Eighteen studies (36%) analysed baseline or follow-up data of different cohort studies.

About two-thirds of the included studies (n=35) recruited their HIV infected samples from clinical settings.[18–52] The rest obtained their samples from the general population through household surveys and community samples. Additionally, many of the studies included samples of PLWH50 + lower than 100 (n=*22*). Overall, the samples of PLWH50 + per study ranged from 19 to 1048,[36 53] while those for HIV uninfected older adults ranged from 17 to 4022.[22 54]

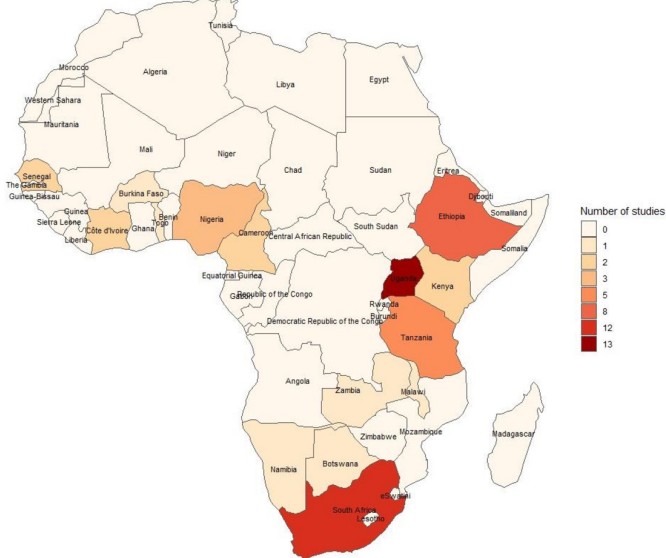

**Figure 2** Geographical distribution of the included studies within SSA. SSA, sub-Saharan Africa.

Common HIV disease markers such as the duration of HIV infection, current/nadir CD4 +T cell count, duration of ART use, viral suppression level among PLWH50 + were reported in a few studies. However, there was extensive heterogeneity in how the information was reported and the level of detail given. In 42% of the included studies, all PLWH50 + were on ART. Six other studies included only ART naïve PLWH50+. Among ART users, viral suppression was reported in five studies where it was documented to range from 73% to 90%. Nadir CD4 +T cell count among PLWH50 + was reported in four studies where most of the participants (about 60%) had a count lower than 200. The duration of HIV infection among PLWH50 + ranged from 3.5 to 12 years (among the few studies that reported this variable).

Only eight studies examined the health outcomes of interest using an exclusively HIV infected older sample.[25 33 40 45–49] Eighteen other studies used HIV uninfected older adults as comparison groups, while the rest included HIV infected young adults (the ages varied extensively).

## Depression and anxiety

Depression was reported in 25 studies, while anxiety was reported in two studies. Notably, depression was assessed using multiple scales with diverse scoring systems. Six of the studies used the 9-item Patient Health Questionnaire (PHQ-9), while a similar number used the Centre for Epidemiological Studies Depression scale. Five other studies used the Mini International Neuropsychiatric Interview (MINI), while the rest utilised a host of different scales. On the other hand, anxiety was assessed using the Schedule for Clinical Assessment in Psychiatry and from clinical records.[30 39] Nine of the depression studies[19 29 44 49 51 55–58] reported major depressive disorders (MDD), while the remaining sixteen studies reported the prevalence of depressive symptoms. To synthesise and report the available evidence, depression (whenever used in this review) captures both the depressive symptoms (derived from brief screening measures) and Major Depressive Disorder (based on structured diagnostic interviews, for example, the MINI by specialised professionals or trained non-specialised workers). More information on this outcome is presented in the online supplemental table 1.

## Prevalence of depression and anxiety

Overall, the prevalence of depression ranged from 6% to 59%. We found no clear pattern of over-reporting or under-reporting of depression prevalence in the included studies, considering the different cut-offs used. For instance, in studies using the PHQ-9, the prevalence of depression ranged from 47% to 49% (with a cut-off of ≥5) and 16% to 45% when using a cut-off of ≥10. We also noted a similar observation when we took the ART status of PLWH into account. Specifically, the prevalence of depression ranged from 8% to 59% among studies whose participants were entirely on ART compared with

6% to 54% among studies with mixed or naïve PLWH. However, studies that used structured interviews such as the MINI, Composite International Diagnostic Interview (CIDI) and Diagnostic and Statistical Manual of Mental Disorders, fourth edition (DSM-IV) to assess for depression were noted to frequently report lower prevalence of depression among PLWH50+ (ranging from 6% to 30%)[44 57] compared with studies that used brief screening scales such as the PHQ-9 (prevalence ranging from 8% to 59%).[27 59]

Of note, eight studies documented higher prevalence of depression among PLWH50 + than their comparison groups.[20 23 27 36 37 43 55 56] On the contrary, 11 studies reported a lower prevalence of depression among PLWH50 + than their comparison groups.[19 24 28 29 31 32 44 50 51 58 59] In comparison, three other studies reported minimal or no differences in depression between PLWH50 + and their comparison groups.[34 42 57] In one population-based study conducted in rural South Africa, neither diagnosis nor treatment of HIV was significantly associated with depression when compared with being uninfected.[54]

Anxiety was reported in two studies.[30 39] In one of them,[39] PLWH50 + on ART presented with the same proportion of anxiety symptoms with HIV uninfected older adults at 3%. In the other study,[30] the prevalence of anxiety disorders among PLWH50 + on ART was 21% compared with 22% in HIV infected young adults (18–50 years).

## Reported correlates of depression

Correlates of depression among PLWH50 + were reported only in eight studies.[27 28 46 48 51 54 55 58] Among sociodemographic variables, older age,[27 54] declining socioeconomic status,[55] being unemployed,[46] being female,[28 58] HIV status disclosure[48] and urban residency[58] were associated with an elevated risk of depression while being married, employed (full-time or part-time work),[54] old age,[51] higher educational attainment[48] and belonging to the highest wealth quintile[28] were associated with reduced levels of depression. However, in some studies, household wealth quintile, education levels,[51] and age[25] were not significant correlates of depression. In other studies, being a current/former smoker,[46] increasing disability scores, decreasing mean handgrip strength, reported back pain, not having hypertension,[55] HIV stigma[48] and caregiving (for adult children)[58] were significantly associated with elevated levels of depression. On the other hand, receiving a government grant[58] and resilience[48] were significantly associated with reduced levels of depression. Details of this data are presented in online supplemental table 1.

## Cognitive and neurological functioning

These outcomes were reported by 13 studies, as highlighted in table 1. Of these, five studies quantified the prevalence of HIV-associated neurocognitive disorders (HAND) using different measures,[25 35 38 40 49] two studies reported findings on dementia[21 60] while five studies documented findings on cognitive function/

**Table 1** Summary of prevalence estimates and correlates for cognitive impairment/neurological impairment

**Studies reporting the prevalence of HIV-associated neurocognitive disorders (HAND) or cognitive impairment**

| Author, publication year and country | Study design | Sample size | Treatment status of PLWH | Assessment tool used | Cut-off score or diagnosis criteria | Information on local tool validation | Prevalence estimates/ other results for PLWH50+ | Correlates reported |
|---|---|---|---|---|---|---|---|---|
| Kellett-Wright, (2020); Tanzania[49] | Cross-sectional | 235 PLWH50+ | 95.5% on ART | International HIV Dementia Scale (IHDS) and (Six Item Dementia Screen, IDEA) | Frascati criteria with a detailed consensus panel discussion | Diagnostic accuracy (area under the receiver operating characteristic curve: 0.64–0.67 for IHDS and 0.65–0.71 for IDEA | Prevalence of symptomatic HAND (MND and HAD) was 25.3%. | NR |
| Eaton, (2020); Tanzania[25] | Cross-sectional | 253 PLWH50+ | 94.8% on ART | Comprehensive NP battery | Frascati criteria for classifying HAND | NR | Prevalence of symptomatic HAND (HAD and MND) was 21.7%. | PLWH were at higher risk of having symptomatic HAND if they lived alone(OR 2.6), were illiterate (OR 3.2) or older at the time of HIV diagnosis (OR 1.1). HIV-specific factors were not related to symptomatic HAND. Vascular risk factors, such as smoking status, were not associated with symptomatic HAND. |
| Oumar, (2020); Burkina Faso[40] | Cross-sectional | 102 PLWH50+ | All on ART | Mini Mental State Examination (MMSE) | Frascati criteria for classifying HAND | NR | 23.5% had neurocognitive disorders | In a multivariate analysis, age (OR 4.6) and educational status (OR 2.6) were associated with neurocognitive disorders. |

Continued

**Table 1** Continued

| Author, publication year and country | Study design | Sample size | Treatment status of PLWH | Assessment tool used | Cut-off score or diagnosis criteria | Information on local tool validation | Prevalence estimates/ other results for PLWH50+ | Correlates reported |
|---|---|---|---|---|---|---|---|---|
| Mugendi, (2019); Kenya[38] | Cross-sectional | 64 PLWH50+; 281 PLWH (20–49 years) | All on ART | Montreal Cognitive Assessment (MoCA) and IHDS | Combined scores of the two tools: a score of ≤26 on MoCA and ≤10 on the IHDS. | NR | Among PLWH50+, symptomatic HAND was 19.2% | Reported correlates not aggregated by age. |
| Tsegaw, (2017); Ethiopia[35] | Cross-sectional | 67PLWH56+; 526 PLWH (18–55 years) | All on ART | IHDS | ≤9.5 | NR | 61.2% among PLWH56 + had HAND | Reported correlates not aggregated by age. |
| Kobayashi, (2019); South Africa[53] | Population survey | 1048 PLWH40+; 4011 HIV uninfected older adults ≥40 years | 63.9% on ART | Brief cognitive battery (orientation, immediate and delayed recall). | ≤1.5 SDs below the mean composite time orientation and memory score. | Locally adapted | 4.0% of PLWH40 + had cognitive impairment compared with 9.0% of uninfected older adults ≥40 years | Independent predictors of lower cognitive score were being older, female, unmarried, not working, having low formal education, low socioeconomic status, and having a history of cardiovascular conditions. |
| Ssonko, (2018); Uganda[33] | Cross-sectional | 411 PLWH50+ | All on ART | Folstein's MMSE | NR | NR | Mild cognitive impairment was associated with falls in the previous 12 months. | NR |

**Studies reporting NP test scores**

Continued

**Table 1** Continued

| Author, publication year and country | Study design | Sample size | Treatment status of PLWH | Assessment tool used | Cut-off score or diagnosis criteria | Information on local tool validation | Prevalence estimates/ other results for PLWH50+ | Correlates reported |
|---|---|---|---|---|---|---|---|---|
| Asiimwe, (2020); South Africa[61] | Population survey | 1048 PLWH40+; 3512 HIV uninfected older adults ≥40 years | 68.6% on ART | A battery of conventional instruments and the Oxford Cognitive Screen-Plus (OCS-Plus) | Summarised mean cognitive scores between participants with and without HIV on the two measures. | Previously validated in the study setting | PLWH scored on average 0.06 (95% CI 0.01 to 0.12) SD units higher on the conventional cognitive function measure and 0.02 (95% CI −0.07 to 0.04) SD units lower on the OCS-Plus measure than HIV-negative participants. | ART use was non-significantly associated with better cognitive outcomes among PLWH in both instruments. |
| Moucheraud, (2020); Malawi[51] | Cross-sectional | 74 PLWH50 + and 60 young PLWH (30–49) years | All on ART | Brief cognitive battery (MMSE) | NR | NR | Mean cognitive scores: 16.0 among PLWH50 + vs 17.4 among young PLWH. | NR |
| Cassimjee, (2017); South Africa[22] | Cross-sectional | 33 PLWH50+; 17 HIV uninfected older adults (mean age 52.2) | NR | Dementia Rating Scale-2, the Stroop Colour and Word Test, the Symbol Digits Test and the D-KEFS Trail Making Test. | N/A | NR | PLWH, in comparison to the uninfected group, had significantly poorer performance profiles in global cognitive functioning, memory, executive functioning, psychomotor functioning, and processing speed. | NR |

**Studies reporting dementia**

**Table 1** Continued

| Author, publication year and country | Study design | Sample size | Treatment status of PLWH | Assessment tool used | Cut-off score or diagnosis criteria | Information on local tool validation | Prevalence estimates/ other results for PLWH50+ | Correlates reported |
|---|---|---|---|---|---|---|---|---|
| Joska, (2019); South Africa[60] | Population survey | 55 PLWH50+; 1095 HIV uninfected older adults ≥50 years | 86.1% on ART | Brief Community Screening Instrument for Dementia | NR | Previously validated | 18.2% of PLWH50 + had dementia compared with 10.7% of the uninfected older adults ≥50 years | Dementia status was associated with HIV infection (OR 2.2, 95% CI 1 to 4.5), older age (OR 1.0, 95% CI 1.0 to 1.1) and depressive symptoms (OR 2.7, 95% CI 1.8 to 3.9). |
| Atashili, (2013); Cameroon[21] | Cross-sectional | 400 HIV infected adults (20–53 years old) | All on ART | IHDS | ≤10 on IHDS | Not validated | The odds of having a positive dementia screen increased progressively with increasing participants' age, | Not aggregated by age |
| **Studies reporting HIV-associated neuropathy** | | | | | | | | |
| Obimakinde, (2020); Nigeria[39] | Cross-sectional | 62 PLWH60+; 162 HIV uninfected older adults ≥60 years | All on ART | Extracted from participants medical notes | Clinical notes | NR | 19.0% in PLWH60 + had peripheral neuropathy | NR |

ART, antiretroviral treatment; D-KEFS, Delis-Kaplan executive function system; HAD, HIV-associated dementia; MND, mild neurocognitive disorder; NA, not applicable; NP, neuropsychological; NR, not reported; PLWH50+, people living with HIV ≥50 years old.

impairment.[22 33 51 53 61] The remaining study reported the proportion of HIV-associated peripheral neuropathy among PLWH50+.[39]

## Prevalence estimates of reported outcomes

The prevalence of symptomatic HAND among PLWH50 + on ART ranged from 19%[38] to 61%.[35] Two of the HAND studies documented elevated proportions of symptomatic HAND among PLWH50 + on ART compared with their younger counterparts.[35 38] In rural South Africa, Joska et al examined the prevalence of dementia and its associations in a population survey of 1150 rural elderly participants. They found no significant differences in the prevalence of dementia between PLWH50+ (86% on ART) and HIV uninfected counterparts 18% vs 11%, respectively.[60] In a different study in rural South Africa, a baseline analysis of the Health and Ageing in Africa—A Longitudinal Study of an INDEPTH Community (HAALSI), HIV infected older adults (64% on ART) registered a lower prevalence of cognitive impairment (4%) than their uninfected peers (9%).[53] In a separate analysis of the same baseline data of the HAALSI cohort,[61] HIV infected older adults scored on average 0.06 (95% CI 0.01 to 0.12) SD units higher on a conventional cognitive function measure and 0.02 (95% CI − 0.07 to 0.04) SD units lower when using the Oxford Cognitive Screen-Plus (OCS-Plus) measure than their HIV-negative counterparts.

## Correlates of cognitive function/impairment

The correlates of cognitive function/impairment were reported in seven studies. The key predictors of reduced cognitive function were mainly sociodemographic: older age (≥50 years),[21 40 53 60] living alone,[25] being illiterate or having low formal education level,[25 53] being older at the time of HIV diagnosis,[25] being female, unmarried and having low socioeconomic status[53] were associated with a higher risk of impairment. However, in some studies, the effects of age, education and marital status were not significant.[25 60] Other correlates of cognitive impairment included the presence of depressive symptoms,[60] a history of cardiovascular conditions[53] and a history of falls in the past 12 months.[33] HIV specific factors (such as nadir CD4 count, ART use)[25 61] and vascular risk factors (such as smoking status, body mass index and blood pressure)[25] were not associated with cognitive impairment.

## Frailty/grip strength
### Prevalence estimates

Frailty was reported in three studies.[33 47 62] In the first study, a hospital-based cross-sectional study of 145 PLWH50 + on ART in Tanzania, frailty assessment was completed using three measures (the Fried frailty phenotype (FFP), the Clinical Frailty Scale (CFS) and Brief Frailty Instrument for Tanzania (B-FIT 2)).[47] In this study, the authors reported low levels of frailty: 3% (when using the FFP method) and 1% (when

using the CFS and BFIT-2 methods). The second study documented a similar prevalence of frailty among 292 PLWH50+ (ART status not reported) and 322 HIV uninfected peers in rural South Africa (15% vs 18%, respectively), using a modified version of the FFP.[62] In the third study, increasing frailty scores were significantly associated with increasing polypharmacy levels in a sample of 411 PLWH50 + on ART in Uganda.[33] Five other studies[18 45 57 63 64] reported mean handgrip strength as an indicator of frailty, as shown in table 2.

## Reported correlates of frailty

The correlates of frailty were reported in two studies. Being female, unmarried[47] and older[47 62] were significantly associated with worse frailty scores, and there were no associations between inflammatory biomarkers with frailty.[62] Factors associated with mean handgrip strength were reported in four of the studies. In one, HIV status was significantly associated with weaker handgrip strength.[57] In another study, being female and divorced/widowed were significantly associated with weaker handgrip strength, while having a secondary education level was significantly associated with better handgrip strength. Table 2 gives details of this outcome.

## Health-related quality of life

This outcome was reported in eight studies. Three-quarters of the studies used the WHO Quality of Life measure to assess HRQOL. The overall mean quality of life scores (out of 100) ranged from 42 to 83 among PLWH50 + on ART. In two studies, PLWH50+ (partially on ART) reported better quality of life scores than their uninfected counterparts.[63 65] In contrast, four other studies reported a lower quality of life scores[39 41 52 66] among PLWH50 + compared with to HIV uninfected older adults or HIV infected young adults.

The correlates of quality of life were reported in only one study,[65] where having some source of income and being in the upper two wealth quintiles were significantly associated with better quality of life among PLWH50+. Additionally, being female and previously married were associated with reduced odds of good quality of life in PLWH50+. Table 3 gives more details on this outcome.

Overall, a significant number of the included studies (about 35%) had a low quality score (≤2 – indicating a high risk of bias).[19–22 26 29 31 35–41 48 51 66] Specifically, about two-thirds of the included studies did not report participant response rates. Sixteen studies used non-probability sampling methods or did not report their sampling methods. Close to half of the included studies had substantially small samples of PLWH50+ (≤100). Furthermore, a substantial number of studies (about 40%) provided scarce information regarding their measures' reliability and validity. Details of the quality assessment of all included studies are captured in online supplemental table 2.

**Table 2** Summary of prevalence estimates and correlates for frailty and handgrip strength

| Author, publication year and country | Study design | Sample size | Treatment status of PLWH | Assessment tool used | Cut-off score | Tool validation | Prevalence estimates/ other results for PLWH50+ | Correlates reported |
|---|---|---|---|---|---|---|---|---|
| **Studies that assessed frailty comprehensively** | | | | | | | | |
| Bristow, (2021); Tanzania[47] | Cross-sectional | 145 PLWH50+ | 99.3% on ART | Fried frailty phenotype (FFP), Clinical Frailty Scale (CFS) and B-FIT 2. | FFP: A score of 1–2 was prefrail, and a score of ≥3 was frail; B-FIT (A score of 8/20); CFS (a score of 5–9) | B-FIT and FFP previously validated in Tanzania. | 2.8% (95% CI 0.09% to 5.4%) when using the FFP. 0.68% (95% CI −0.66% to 2.04%) when using the CFS or B-FIT 2). | FFP frailty was associated with female gender (p=0.006), marital status (p=0.007) and age (p=0.038). Weight loss was the most common FFP domain failure. |
| Edwards, (2020); South Africa[62] | Cross-sectional | 292 PLWH50 + and 322 HIV uninfected adults (≥50) years. | NR | FFP | Presence of ≥3 of the 5 FFP phenotypes | Adapted in the local population | Frailty was similar for HIV negative and PWH (17.7% vs 14.7%, p=0.72) respectively as was prefrailty status (66.5% vs 63.4%). | The prevalence of frailty increased with age. There was no association between any of the inflammatory biomarkers and frailty and prefrailty. |
| Ssonko, (2018); Uganda[33] | Cross-sectional | 411 PLWH50+ | All on ART | Self-report 40 -item Frailty Index Questionnaire | NR | NR | Frailty Index Scores of 5 to 6 (PR 10.6, 95% CI 1.4 to 78, p=0.02), and seven or more (PR 17.4, 95% CI 2.4 to 126.5, p=0.005) were associated with polypharmacy. | Increasing Frailty score was associated with increasing levels of polypharmacy. |
| **Studies that assessed handgrip strength** | | | | | | | | |
| Bernard, (2020); Co'te d'Ivoire & Senegal[45] | Cross-sectional | 333 PLWH50+ | All on ART | Calibrated Jamar hydraulic hand dynamometer | A performance less to the median obtained in each gender group (male/female) | NR | 43.2% of women had low grip strength. 42.5% of men had low grip strength. | NR |

Continued

**Table 2** Continued

| Author, publication year and country | Study design | Sample size | Treatment status of PLWH | Assessment tool used | Cut-off score | Tool validation | Prevalence estimates/ other results for PLWH50+ | Correlates reported |
|---|---|---|---|---|---|---|---|---|
| Fliteau, (2017); Zambia and Tanzania[18] | Randomised trial | PLWH50+: 148 at baseline; 72 at 12 wks, and 50 at 2–3 years ART | ART naïve at baseline | Digital handgrip dynamometer | NA | NR | Mean grip strength change: 18.7 kgs at baseline, 20.4 kgs at 12 weeks and 23.1 kg at 2–3 years. | Reported correlates not aggregated by age. |
| Mugisha, (2013); Uganda[63] | Cross-sectional | 198 PLWH50+; 311 HIV uninfected adults ≥50 years | 51.0% on ART | Digital handgrip dynamometer | NA | NR | 25.7 among PLWH50+; and 22.1 among the uninfected older adults ≥50 years. | NR |
| Negin, (2012); South Africa[57] | Cross-sectional | 142 PLWH50+; 2722 HIV uninfected older adults ≥50 years | NR | Digital handgrip dynamometer | NA | Locally validated | Mean grip strength in PLWH50+ (33.5 kg) was weaker than HIV-uninfected individuals (38.3 kg). | When controlling for age and sex among those aged 50 years and older, HIV status was significantly associated with weaker grip strength. |
| Scholten, (2011); Uganda[64] | Cross-sectional | 189 PLWH50+; 2017 HIV affected 50+; and 104 HIV uninfected older adults ≥50 years | 51.0% on ART | Digital handgrip dynamometer | NA | Locally adapted | 25.8 among PLWH50+; and 25.3 among HIV affected 50+ and 25.2 among the uninfected older adults ≥50 years. | Women scored poorly than men. Married respondents had significantly better scores than widowed and divorced respondents. Respondents with at least secondary education had significantly better scores. |

ART, antiretroviral treatment; B-FIT 2, Brief Frailty Instrument of Tanzania; NA, not applicable; NR, not reported; PLWH50+, people living with HIV ≥50 years old; PR, prevalence ratio.

**Table 3** Summary of results for health-related quality of life

**Studies that utilised different versions of the WHO Quality-of-Life Scale (WHOQOL)**

| Author, publication year and country | Study design | Sample size | Treatment status of PLWH | Assessment tool used | Cut-off score | Information on local tool validation | Prevalence estimates/other results | Correlates reported |
|---|---|---|---|---|---|---|---|---|
| Obimakinde, (2020); Nigeria[39] | Cross-sectional | 62 PLWH60+; 124 HIV uninfected older adults ≥60 years | All on ART | 8-item WHOQoL | NA | NR | Mean QoL (SD): 82.7 (8.9) among PLWH50+; 86.2 (6.5) in the uninfected adults ≥50 years. | NR |
| Mugisha, (2013); Uganda[63] | Cross-sectional | 198 PLWH50+; 311 HIV uninfected older adults ≥50 years | 51.0% on ART | 8-Item WHOQoL | NA | NR | Mean QoL scores: 57.2 among PLWH50+; and 45.3 in the uninfected older adults ≥50 years | NR |
| Nyirenda, (2013); Uganda and South Africa[70] | Cross-sectional | 400 PLWH50+; 732 HIV uninfected older adults ≥50 years | About 22% on ART | 8-Item WHOQoL | NA | Locally adapted and validated | Older people in South Africa were more likely to have a better quality of life (aOR 2.15, 95% CI 1.60 to 2.90) than in Uganda. | NR |
| Nyirenda, (2012); South Africa[65] | Cross-sectional | 203 PLWH50+; 219 HIV uninfected older adults ≥50 years | 23.7% on ART | 8-Item WHOQoL | NA | Locally adapted | Mean QoL scores: 62.5 among PLWH50+; and 59.4 in the uninfected older adults ≥50 years. | Being female and having been previously married was associated with lower odds of good QoL in PLWH50+ |
| Yaya, (2019); Togo[52] | Cross-sectional | 154 PLWH50+; 726 PLWH (15–49 years) | 90.5% on ART | 31-Item WHOQOL-HIV BREF | ≥77.3 was considered good quality of life. | NR | 72.7% of PLWH50 + had a good global quality of life compared with 77.3% in the younger PLWH. | Correlates not aggregated by age |
| Maniragaba, (2018); Uganda[66] | Cross-sectional | 28 PLWH50+; 884 HIV uninfected older adults ≥60 years | NR | An adapted version of the WHOQOL-OLD | An index (good, fair and poor quality of life) . | NR | PLWH had poor quality of life (OR 0.45; 95% CI 0.22 to 0.93) compared with those who were HIV uninfected. | Not aggregated by HIV status |

Continued

**Table 3** Continued

**Studies that used other measures of quality of life**

| Author, publication year and country | Study design | Sample size | Treatment status of PLWH | Assessment tool used | Cut-off score | Information on local tool validation | Prevalence estimates/other results | Correlates reported |
|---|---|---|---|---|---|---|---|---|
| Parcesepe, (2020); Tanzania[41] | Cross-sectional | 82 PLWH50+; 824 PLWH (18–49 years) | ART naïve | HIV/AIDS-Targeted Quality of Life Scale | NA | Previously validated in other SSA settings | Mean QoL score: 68 among PLWH50+, and 72.4 among PLWH (18–49 years) | Reported correlates not aggregated by age. |
| Harding, (2014); Kenya and Uganda[26] | Cross-sectional | 1337 adults aged 18–70 years (no of PLWH50 + NR) | 49.9% on ART | Medical Outcome Scale -HIV (MOS-HIV) | NA | Previously validated in other SSA setting | Mean Mental Health score: 46.4 among PLWH50+; 46.2 in PLWH (18–49 years) Physical health score: 42.9 among PLWH50+; and 44.8 in PLWH (18–49 years) | Reported correlates not aggregated by age. |

AOR, adjusted OR; ART, antiretroviral treatment; NA, not applicable; NR, not reported; PLWH50+, people living with HIV ≥50 years old; QoL, quality of life; SSA, sub-Saharan Africa.

## DISCUSSION
### Summary of main findings
Our review shows that the research on depression, anxiety, cognitive function, frailty and HRQoL among PLWH50 + in SSA is still scanty and largely concentrated in Uganda, South Africa and Ethiopia. Overall, PLWH50 + presented with wide-ranging prevalence of depression (6% to 59%), cognitive impairment (4% to 61%) and frailty (3%–15%). Additionally, the correlates of these outcomes were rarely investigated. Nonetheless, those reported were mainly sociodemographic variables, many of which were inconsistent. We find the current evidence inadequate to characterise the real burden and determinants of these outcomes in the region, partly because of the limited number of well-designed studies, heterogeneous findings, and substantial methodological limitations observed in many included studies. Therefore, any policies and programmes about the well-being of PLWH50 + in the region have been made on the basis of minimal evidence. The rising numbers of PLWH50 + in SSA and the potentially elevated burden of age-associated comorbidities in this population warrant more research across the region. Such data will provide the evidence needed to develop appropriate policies, programmes and interventions for the successful ageing of PLWH in SSA.

### Depression and anxiety
Depression was the most frequently reported outcome in about half of the included studies. This finding is consistent with the available literature on HIV, which recognises depression as the most commonly occurring psychiatric complication in this population.[67] However, only two studies investigated anxiety, a CMD with a similar prevalence to those seen in depression.[68] The under investigation of anxiety disorders is of particular concern because accumulating research involving PLWH50 + in HICs has documented elevated anxiety levels, as high as 65%.[68] More research is needed on this outcome in the region.

Our review also noted wide-ranging prevalences of depression and anxiety. This finding corroborates previous review findings on the prevalence of CMDs in the region, although in younger HIV populations.[69] The observed variation could reflect significant contextual differences across the SSA region, for example, social environmental milieu, healthcare systems, patterns of the burden of diseases and mental health resources, which are likely to alter the risk profile of these adults.[70] It could also be attributed to potential differences in study populations, study participants and the use of diverse measurement tools (including different cut-offs for the same measures). Indeed, in the current review, depression was assessed using at least 10 different scoring systems with diverse cut-off points. Besides, the majority of the studies did not provide information on the reliability and validity of the tools they used. Hence, it is difficult to draw firm conclusions regarding the reliability, validity and applicability of many of the measures used to report this outcome. The PHQ-9, for instance, was frequently used to report depressive symptoms among the included studies. However, most of the studies relied on previous validation of the tool, the context of which was not given. A recent review examining the validation of brief CMDs screening tools in low-income and middle-income countries did not support the use of PHQ-9 in these settings. It performed poorly in several clinic populations with lower average education but performed very well in high literate populations.[71] This is especially an important consideration in SSA, where older adults are more likely to have low literacy levels. There is a need for rigorous cultural adaptation of these instruments to generate accurate epidemiological data that will inform proper intervention work among PLWH50 + in SSA.

### Hiv-associated cognitive impairment
The neurological involvement in HIV disease progression remains problematic, even in well-controlled PLWH, resulting in poor quality of life. HIV associated cognitive impairment is deemed one of the most prevalent comorbidities in the era of ART. Ageing with HIV would appear to impose a synergistic impact, whereby increased comorbidities, such as hypertension, diabetes or depression, likely contribute to and maintain HIV-associated cognitive impairment. In HICs, PLWH50 + have consistently been shown to have high levels of cognitive impairments, as high as 50%. The present review found a wide-ranging prevalence of HIV-associated cognitive impairment (4%–61%). Our observation is consistent with the available literature, where the prevalence has been noted to vary widely depending on sources used and appears dependent on the populations studied, and approaches used.[72]

Brief cognitive screening tools such as the Mini Mental State Examination and International HIV Dementia Scale (IHDS) were frequently used to document the prevalence of cognitive impairment compared with comprehensive neuropsychological batteries. While the use of the brief screeners in low-resource settings is well understood given the resource gap, researchers should exercise caution on the tools they use. Recent reviews evaluating brief screening tools for neurocognitive impairment in HIV/AIDS have indicated that some of the commonly used tools, for example, the IHDS perform poorly, especially in screening milder cognitive impairments.[73 74] In the current review, information on tool reliability or validity was rarely reported. One of the studies reported a limited diagnostic accuracy of the IHDS. However, the OCS-Plus, one of the most recent screening tools, is promising. So far, the tool has only been used in South Africa among older adults living with HIV, and its initial evaluation yielded excellent construct and external validity.[75] This tool has the potential to address some of the unique challenges and gaps facing resource-limited settings in screening for HAND, including difficulty in performing long test batteries, limited screening tools and a shortage of clinical staff.

## Frailty

Only two studies quantified the prevalence of frailty among PLWH50+, ranging from 3% to 15%. One of the studies was exclusively conducted among PLWH50+, yielding a low frequency of frailty (3%). In the other study, conducted in rural South Africa, the prevalence of frailty was similar between PLWH50 + and their uninfected counterparts (15% vs 18%, respectively). This finding supports the growing evidence in SSA, which suggest that older on effective ART have similar or better health profiles than their uninfected counterparts, contrary to what has been observed in HICs. At the global stage, the frailty prevalence in SSA appears relatively lower than that of HICs which ranges from 5% to 29%.[76] The population reaching older age in many SSA settings may be relatively fit/active, still working, which may be true of PLWH, potentially favouring better physical health in these individuals. Nonetheless, given the limited number of studies on this outcome in SSA, these findings remain preliminary and inconclusive and highlight the urgent need for more studies to shed more light on these observations.

## Health-related quality of life

For some time, HRQoL has emerged as a salient indicator of HIV care and an essential target for HIV-related research, aiming to expand the service paradigm continuum.[77] This new quality of life frontier has been proposed as the 'fourth 90' to the original '90-90-90' testing and treatment target championed by UNAIDS: to ensure that 90% of people with viral load suppression have good HRQoL.[77] In the current review, research on HRQoL among PLWH50 + was scanty. Thus, little is known on the risk and protective factors of patient-reported HRQoL. Further studies should explore the independent effects of aging-related conditions such as CMDs, cognitive impairments and frailty on HRQoL among PLWH50 + in SSA to support healthy ageing even in the context of a chronic illness such as HIV infection.

Despite the observed limitations, the evidence of better mental health,[28 44 50 51 54 58 59] as well as cognition,[53 61] frailty/handgrip strength[47 62 63] and quality of life,[65] among PLWH50 + on ART (mainly among the robust studies with low risk of bias) compared with uninfected older adults in the current review was striking. This observation could be part of an emerging body of evidence from SSA showing that the mental and well-being outcomes among PLWH are comparable to those without HIV. The mechanisms responsible for this observation are not well established and require further examination across many settings in SSA. However, it could be attributed to a few plausible reasons, including (1) the direct impact of ART on viral replication, inflammatory pathways and downstream disease pathophysiology, the so-called 'ART advantage',[78 79] (2) a survivorship bias among PLWH50 + who access and remain in HIV care and (3) beneficial spillover effects of ART programmes such as enhanced access to and utilisation of primary care services resulting in early identification and management of comorbid conditions.[79–81] This observation could also be related to the fact that PLWH50 + on ART are more likely to use biomedical facilities due to better education and higher socioeconomic status than their HIV uninfected peers. More studies, preferably longitudinal ones, will shed more light on this observation. As it stands, however, this finding offers an interesting contrast to the commonly reported findings from HICs, which generally show that PLWH50 + present with worse physical and mental health outcomes compared with age-matched HIV uninfected people.[5 68 82] These findings may suggest context-specific differences in the drivers of physical and mental outcomes, including the demographic and risk factor profile as well as the role of engagement in HIV care—among those with and without HIV, especially when comparing low-income and middle-income to HICs. Future studies should explore the extent to which HIV programmes could be promoting the well-being of PLWH50 + in SSA compared with HIV uninfected older adults in the general population who are likely to bear the brunt of the prevailing poor health systems and primary care systems in the region.

## The correlates of mental and well-being outcomes

The correlates of CMDs, cognitive impairment, frailty and HRQoL have been scarcely investigated in SSA. In the current review, the frequently reported correlates were sociodemographic in nature, many of which (eg, age, formal education levels and wealth status) were inconsistent. However, being female was consistently associated with declining mental health, cognitive function, HRQoL and frailty status. This finding mirrors the results of a recent review of older women's physical and psychological health residing in SSA, which indicated that women were significantly more likely to report poor health compared with men.[83] This finding highlights the need for adequate representation of both sexes in research studies, incorporating a gender lens in their analyses to understand better the unique health challenges that each gender faces. Further studies are also urgently needed in SSA to identify potential risk and protective factors, especially those that could be modified, to suggest how to best integrate the management of these impairments in HIV care guidelines in SSA. Future studies should explore the influence of HIV-related factors, psychosocial risk (eg, loneliness, ageism, stigma, social support, living arrangements) and other comorbidities on the health and well-being of PLWH50+. To promote healthy ageing in the context of chronic conditions, such as HIV infection, the psychological and social domains are especially critical as they could be a way to compensate for physiological limitations (ie, multiple chronic conditions) and allow even in the context of disease and disability, to experience optimal quality of life.

## Strengths and limitations

To our knowledge, this is the first comprehensive review of CMDs, cognitive impairment, frailty and HRQoL

among PLWH50 + living in SSA. Our expanded search of five online databases and snowballing techniques minimised the risk of missing relevant studies. However, our review should be considered in the context of important limitations. Despite a thorough search strategy, some articles may have been missed, especially in excluding non-English language studies. However, these were very few. The substantial amount of heterogeneity across the studies in terms of, for example, study design, operationalisation of age, definitions of outcomes and measures made comparisons difficult and did not allow a meta-analysis to be done. Furthermore, most of the studies recruited PLWH50 + from HIV clinics whose attendees may represent the less-healthy end of the spectrum of service users leading to overestimation of reported outcomes. Conversely, those at the filter end of the cohort may be more able to attend or be proactive concerning their well-being, leading to underestimation of outcomes. Besides, the substantial methodological limitations observed might have biased the reported findings, hence limiting their external validity.

## CONCLUSIONS

Overall, the existing research on the burden and determinants of mental and well-being outcomes among PLWH50 + in SSA is scantly addressed. We also noted substantial limitations in many of the included studies, which need to be addressed in future studies. Our findings underscore the need to increase the evidence base of these outcomes in the region. Future studies should: (1) explicitly target older persons ≥50 years living with HIV, (2) carefully select a comparison group and have sufficient sample sizes for adequate statistical testing, (3) expand the geographical distribution of these studies in many SSA settings and use adequately standardised survey tools adapted for the region and (4) generate more data on the risk and determinants of these outcomes using a gender lens. There is also an ongoing need for well-designed longitudinal cohort studies to study the temporal patterns of these outcomes in the region.

**Acknowledgements** The authors would like to thank the Director of Kenya Medical Research Institute for granting permission to publish this work.

**Contributors** PNM and AAA conceptualised and designed the review. PNM and AM reviewed titles, abstracts and full-text papers for eligibility as well as data extraction. PNM wrote the first draft of the manuscript. AM, CRJCN, RW and AAA reviewed and edited the prepared manuscript.

**Funding** This work was funded by the Wellcome Trust International Master's Fellowship to PNM (Grant number 208283/Z/17/Z). Further funding supporting this work was from (1) the Medical Research Council (Grant number MR/M025454/1) to AAA. This award is jointly funded by the UK Medical Research Council (MRC) and the UK Department for International Development (DFID) under MRC/DFID concordant agreement and is also part of the EDCTP2 programme supported by the European Union; (2) DELTAS Africa Initiative (DEL-15–003). The DELTAS Africa Initiative is an independent funding scheme of the African Academy of Sciences (AAS)'s Alliance for Accelerating Excellence in Science in Africa (AESA) and supported by the New Partnership for Africa's Development Planning and Coordinating Agency (NEPAD Agency) with funding from the Wellcome Trust (107769/Z/10/Z) and the UK government. For the purpose of Open Access, the

author has applied a CC-BY public copyright licence to any accepted manuscript version arising from this submission.

**Disclaimer** The funders did not have a role in the design and conduct of the study or interpretation of study findings. The views expressed in this publication are those of the author(s) and not necessarily those of AAS, NEPAD Agency, Wellcome Trust or the UK government.

**Map disclaimer** The inclusion of any map (including the depiction of any boundaries therein), or of any geographic or locational reference, does not imply the expression of any opinion whatsoever on the part of BMJ concerning the legal status of any country, territory, jurisdiction or area or of its authorities. Any such expression remains solely that of the relevant source and is not endorsed by BMJ. Maps are provided without any warranty of any kind, either express or implied.

**Competing interests** None declared.

**Patient consent for publication** Not applicable.

**Ethics approval** This work is part of a PhD project to the first author. The project has obtained ethical clearance from the Kenya Medical Research Institute-Scientific and Ethics Review Unit (KEMRI/SERU/CGMR-C/152/3804).

**Provenance and peer review** Not commissioned; externally peer reviewed.

**Data availability statement** All data relevant to the study are included in the article or uploaded as online supplemental information. All data underlying the results are available as part of the article, and no additional source data are required.

**ORCID iD**
Patrick Nzivo Mwangala http://orcid.org/0000-0001-9046-1465

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
