## [Reviewer comments · BMJ Open]

ARTICLE DETAILS

TITLE (PROVISIONAL)	Mental health and wellbeing of older adults living with HIV in sub-Saharan Africa: a systematic review
AUTHORS	Mwangala, Patrick; Mabrouk, Adam; Wagner, Ryan; Newton, Charles; Abubakar, Amina A.

VERSION 1 – REVIEW

REVIEWER	David van de Vijver Erasmus MC, Viroscience
REVIEW RETURNED	17-May-2021

GENERAL COMMENTS	Mwangala and colleagues performed a systematic review on empirical evidence on common mental disorders, cognitive impairment, frailty, and health-related quality of life among people living with HIV who are aged at least 50 years residing in sub-Saharan Africa. The systematic review focused on the prevalence and correlates of symptoms indicating a mental disorder. Although there are clear indications that frailty and mental disorders are more common among older people living with HIV, no studies systematically reviewed the literature on this topic in sub-Saharan Africa. The paper reports a wide range in the prevalence and correlates of mental disorders, cognitive impairment, frailty, and health-related quality of life. The paper also reports that correlates of mental disorders were rarely investigated and frequently inconsistent. Major comment I have one major comment: the studies included in the review used several scales to classify mental health problems. For instance, depression is measured using at least ten different scoring systems. In addition, the studies included in the review used different cut-off values to classify depressive symptoms. One system that was used by many studies was the 9-item patient health questionnaire, of which some studies used a cut-off of 5 and other studies a cut-off of 10 to classify depressive symptoms. A cut-off of 5, however, includes milder forms of depression as compared to a cut-off of 10. Similarly, some scales studies major depression as an outcome. The severity of depression has an influence on the outcome. I therefore recommend the authors to analyse (or group) the data also by the scale and cut-off that is used. The authors should also discuss the merits and validity of the different scales that were used in reporting mental disorders.
---

REVIEWER	Charlotte Bernard University of Bordeaux
REVIEW RETURNED	22-Jun-2021

GENERAL COMMENTS

In this review, the authors present results on mental health and wellbeing in older adults living with HIV in SSA. This is a huge work to detail the different information. This topic is important. The paper needs major revisions to clarify some points and present more clearly the results.

1. Please add "mental" in the title

2. Background: reorganize the introduction (for example prevalence, specificities of aging with HIV, marginalized population, scarce data, aims).

3. Methods:

For prevalence, information from studies including middle-aged patients should be removed. Please focus on the 50+. This is the originality of the paper.

For determinants, papers with middle-aged patients but studying the impact of age must be included.

This point is important to clarify. The aim is to have older PLWH50+ but in the method, it is mentioned "involved older PLWH whose mean/median age was ≥ 50 years or studies that aggregated the outcomes of interest by age (including the ≥ 50 age category)" it is confusing. Some studies included only 50+.

"We excluded studies that did not aggregate the outcomes of interest by age and HIV status": not clear what that means

4. Results:

When results are described, it is important to group papers where PLWH are on ART vs mixed or naïve PLWH. Is it necessary to include studies with naïve PLWH? Recommendations are now far from that.

Please organize the paragraph with subtitles: prevalence, associated factors.

Tables need to be clarified and simplified.

Table 1 not necessary. Information from Table 1 needs to be added to the other tables.

ART regimen, sample source, sampling method, comparison group sample are not necessary in the table.

Please reorganize tables. For example for depression: group the data depending on assessment tool (ex: MINI, DSM, clinical evaluation, CESD....) (cut off can be added and info on validation might be too). Mention only the prevalence for 50+. Classify the publication by date publication.

Authors need to reorganize the others tables in the same way.

Is necessary to keep well-being in this paper? There is a lot of information. Maybe it could be presented in another paper?

In Obimakinde, age 60+ not 50+

How the authors compute the prevalence when they had middle-aged patients? It is not clear.

5. Discussion/ conclusions

The conclusion repeats in part the first paragraph of the discussion. Discussion may be reorganized in parts: depression/anxiety, cognitive impairment, frailty to facilitate the reading.

REVIEWER	Makandwe Nyirenda South African Medical Research Council, Burden of Disease Research Unit
REVIEW RETURNED	23-Jun-2021

GENERAL COMMENTS	This is a very important paper in an area with limited empirical evidence. The paper clearly defined the study objectives and methodology. The study results and discussion are also clearly presented. This paper makes a valuable contribution on the health of HIV-infected and uninfected older adults in sub-Saharan Africa. My minor queries for the authors are:  1) In line 33 on page 6, you say 50 studies met the eligibility criteria and were include. However, in lines 34-39 the count of studies and their geographical location suggests more than 50 studies. Were some studies duplicated or multi-country studies? Please clarify. 2) What effect, if any, would the inclusion in the search terms of the full name for HIV, i.e Human Immunodeficiency Virus, have made? 3) The authors need to acknowledge the limitation of excluding non-English language studies. Some crucial studies from Franco-phone sub-Saharan African countries may have been missed and hence results presented biased to that extent.
--

VERSION 1 – AUTHOR RESPONSE

Reviewer 1: Dr. David van de Vijver, Erasmus MC

Mwangala and colleagues performed a systematic review on empirical evidence on common mental disorders, cognitive impairment, frailty, and health-related quality of life among people living with HIV who are aged at least 50 years residing in sub-Saharan Africa. The systematic review focused on the prevalence and correlates of symptoms indicating a mental disorder. Although there are clear indications that frailty and mental disorders are more common among older people living with HIV, no studies systematically reviewed the literature on this topic in sub-Saharan Africa.

The paper reports a wide range in the prevalence and correlates of mental disorders, cognitive impairment, frailty, and health-related quality of life. The paper also reports that correlates of mental disorders were rarely investigated and frequently inconsistent.

Major comment

I have one major comment: the studies included in the review used several scales to classify mental health problems. For instance, depression is measured using at least ten different scoring systems. In addition, the studies included in the review used different cut-off values to classify depressive symptoms. One system that was used by many studies was the 9-item patient health questionnaire, of which some studies used a cut-off of 5 and other studies a cut-off of 10 to classify depressive symptoms. A cut-off of 5, however, includes milder forms of depression as compared to a cut-off of 10. Similarly, some scales studies major depression as an outcome. The severity of depression has an influence on the outcome. I therefore recommend the authors to analyse (or group) the data also by the scale and cut-off that is used. The authors should also discuss the merits and validity of the different scales that were used in reporting mental disorders.

Response:

We appreciate the reviewer for the important feedback. In our revision, we have done the following:

- a) Reorganized Table 1 (previously table 2) 'Summary of prevalence estimates and correlates for common mental disorders.' Specifically, we have grouped the studies according to the scale they utilized (e.g. PHQ-9, CES-D, MINI, geriatric depression scale, CIDI and others). Under each scale categorization, we have also listed the studies according to the cut-offs used to report the outcome. However, we note that the latter was quite challenging (e.g. for studies listed under the CES-D scale) given that the studies used quite diverse cut-offs. [Page 20 to 23 of the main manuscript document]
- b) We have closely examined the potential implications of using different scales in reporting the prevalence of common mental disorders. We did not identify clear patterns of either over-reporting or under-reporting of depressive symptoms when using the different cut-offs, e.g. for the studies using PHQ-9. For the CES-D scale, virtually all the studies used different cut-offs to report depressive symptoms; hence difficult to assess for this. However, studies that used comprehensive or structured interviews such as the MINI, CIDI, DSM-IV to assess for depression were noted to frequently report lower prevalence of depression than those that used brief screening scales such as the PHQ-9. This specific observation is highlighted on page 7 – first paragraph.
- c) We have also reorganized the other tables (Table 2 - reporting cognitive function), Table 3 - reporting frailty) and Table 4 - reporting quality of life) to make them clearer. Table 4 was reorganized by the measures used while Table 2 and 3 were reorganized by the outcomes reported because the studies used quite diverse assessment tools. [Page 24 to 28 of the main manuscript document]
- d) As suggested, we have also discussed the merits and validity of the different scales used to report mental disorders and the other outcomes as a whole. The following excerpts highlight this change in the discussion section:
"in the current review, depression was assessed using at least 10 different scoring systems with diverse cut-off points. Besides, the majority of the studies did not provide information on the reliability and validity of the tools they used. Hence, it is difficult to draw firm conclusions regarding the reliability, validity, and applicability of many of the measures used to report this outcome. The PHQ-9, for instance, was frequently used to report depressive symptoms among the included studies. However, most of the studies relied on previous validation of the tool, the context of which was not given. A recent review examining the validation of brief CMDs screening tools in low and middle-income countries did not support the use of PHQ-9 in these settings. It performed poorly in several clinic populations with lower average education but performed very well in high literate populations [71]. This is especially an important consideration in SSA, where older adults are more likely to have low literacy levels. There is a need for rigorous cultural adaptation of these instruments to generate accurate epidemiological data that will inform proper intervention work among PLWH50+ in SSA." [Page 11 first paragraph]

"Brief cognitive screening tools such as the Mini Mental State Examination and International HIV Dementia Scale (IHDS) were frequently used to document the prevalence of cognitive impairment compared to comprehensive neuropsychological batteries. While the use of the brief screeners in low resource settings is well understood given the resource gap, researchers should exercise caution on the tools they use. Recent reviews evaluating brief screening tools for neurocognitive impairment in HIV/AIDS have indicated that some of the commonly utilized tools, e.g. the IHDS perform poorly, especially in screening milder cognitive impairments [78, 79]. In the current review, information on tool reliability or validity was rarely reported. One of the studies reported a limited diagnostic accuracy of the IHDS. However, the Oxford

Cognitive Screen-Plus (OCSPlus), one of the most recent screening tools, is promising. So far, the tool has only been used in South Africa among older adults living with HIV, and its initial evaluation yielded excellent construct and external validity [80]. This tool has the potential to address some of the unique challenges and gaps facing resource-limited settings in screening for HAND, including difficulty in performing long test batteries, limited screening tools and a shortage of clinical staff. [Page 12 second paragraph]

Reviewer 2: Dr. Charlotte Bernard, University of Bordeaux

In this review, the authors present results on mental health and wellbeing in older adults living with HIV in SSA. This is a huge work to detail the different information. This topic is important. The paper needs major revisions to clarify some points and present more clearly the results.

1. Please add "mental" in the title

Response:

*We have revised the title and captured the suggested addition. The updated title reads '**Mental health and Wellbeing of older adults living with HIV in sub-Saharan Africa: a systematic review.**'* [Page 1]

2. Background: reorganize the introduction (for example prevalence, specificities of aging with HIV, marginalized population, scarce data, aims).

Response:

We appreciate the reviewer for this suggestion. We have reorganized the background as suggested with the following flow: statistics of PLWH50+, specificities of aging with HIV, marginalization of PLWH50+, growing data on aging with HIV in SSA and aims [Page 3 and 4 of the manuscript]

3. Methods:

For prevalence, information from studies including middle-aged patients should be removed. Please focus on the 50+. This is the originality of the paper.

Response:

We appreciate the reviewer for this comment. Generally, HIV and aging is an emergent research area in SSA. As such, the literature on this subject is small (especially on the reported outcomes), though building up. Expectedly, a good number of the available studies have not focused exclusively on older adults living with HIV (those aged at least 50 years) and rarely report findings related to aging or use the recommended age category of ≥ 50 years. However, some of these studies have aggregated their reported outcomes by age, including those now recognized as older adults living with HIV. In our review, we considered these studies that were not exclusively on older adults ≥ 50 years but had aggregated their findings by age (including those at least 50). We also considered studies with samples of middle-aged PLWH but whose participants had a mean or median age of at least 50 years. However, in the former scenario, we note that the samples of older adults ≥ 50 years were more likely to be small. We have noted this observation and recommended that future studies exclusively target PLWH50+ in sufficient numbers to accurately depict the burden of the studied health outcomes in the region. These additional inclusion approaches helped us to

comprehensively capture many of the potential studies in SSA that have reported these outcomes and accurately report their limitations in this initial review in the SSA region.

For determinants, papers with middle-aged patients but studying the impact of age must be included.

This point is important to clarify.

Response:

All the currently reported correlates considered this suggestion. Only studies that included samples of older adults living with HIV ≥50 years and those which studied the impact of age were considered.

The aim is to have older PLWH50+ but in the method, it is mentioned "involved older PLWH whose mean/median age was ≥ 50 years or studies that aggregated the outcomes of interest by age (including the ≥ 50 age category)" it is confusing. Some studies included only 50+.

Response:

We acknowledge this comment. As pointed out in the previous response, the concept of HIV and aging is relatively new in SSA. Thus previous studies (and sometimes current ones) adopt diverse age categorizations to define old age. Also, a good number of previous studies in SSA have not explicitly involved older adults at least 50 years of age. However, they aggregate their findings by age (including the ≥50 age category). Others have included middle-aged adults, but their overall participants' mean or median age is above 50. For instance, the studies emanating from the HAALSI cohort in South Africa and UGANDAC cohort in Uganda have included adults aged ≥40 years, but their participants' mean age is 61.7 and 52, respectively.

The inclusion criteria captured in the aim is clarified in the methods section as follows:

- a) Studies that exclusively involved older adults ≥50 years*
- b) Studies which included PLWH whose mean or median age was ≥50 years*
- c) Studies that aggregated their outcomes of interest by age, including that of ≥50 years.*

*We have further clarified this in the updated review. The following statement has been inserted in the methods section on inclusion criteria: '**Studies were considered eligible if they: (i) involved PLWH50+ or had participants whose mean/median age was ≥ 50 years or aggregated their outcomes of interest by age (including the ≥ 50 age category)...**'*

[Third paragraph of methods section – page 4]

"We excluded studies that did not aggregate the outcomes of interest by age and HIV status": not clear what that means

Response:

*The statement is noted as a specific exclusion criterion. We have revised the statement to make it clearer. Specifically, we have dropped the initial part referring to age as this was a repetition as it is already captured in the earlier statements. The current statement reads, '**We excluded studies that did not aggregate their outcomes of interest by HIV status.**' Here, we excluded studies that did not aggregate their findings by HIV status (among studies that*

had both HIV infected and HIV uninfected participants). [Third paragraph of methods section – page 4]

4. Results:

When results are described, it is important to group papers where PLWH are on ART vs mixed or naïve PLWH.

Response:

We appreciate the reviewer for this comment. We have updated the results section as suggested. The respective changes have been highlighted in yellow in the revised manuscript. Whenever applicable, we have given the estimates for PLWH50+ who are fully on ART versus mixed or ART naïve PLWH50+. However, we did not find any clear pattern of better or worse outcomes by taking this into account. Besides, the individual studies rarely aggregated their outcomes by ART status in instances where there were mixed PLWH50+.

Is it necessary to include studies with naïve PLWH? Recommendations are now far from that.

Response:

We believe it was important to include studies with ART naïve PLWH in the current review to give an accurate picture of the extent of research on these outcomes, given the emerging nature of the subject in the region. Through this approach, we were able to comprehensively examine the available studies and offer informed recommendations regarding the prevailing limitations of the existing literature. Excluding these studies would also mean excluding the studies with mixed groups of PLWH since the individual studies did not aggregate their outcomes by ART status.

Moreover, we also know that a substantial number of PLWH in SSA are not accessing the lifesaving ART. In the latest UNAIDS data, close to 25% of all PLWH in Eastern and Southern Africa were not accessing ART, and the proportion is slightly higher in Western and Central Africa. Many of these participants are especially identified in baseline population surveys, a significant number of whom are OALWH as they are more likely to have a late diagnosis.

Please organize the paragraph with subtitles: prevalence, associated factors.

Response:

We appreciate the review for this important suggestion. Indeed, we have reorganized the paragraphs into the suggested portions. We believe the results are clearer in the revised manuscript. [Pages 6 to 10]

Tables need to be clarified and simplified.

Table 1 not necessary. Information from Table 1 needs to be added to the other tables.

ART regimen, sample source, sampling method, comparison group sample are not necessary in the table.

Response:

We thank the reviewer for this important feedback. We have adopted the suggestions put forward. Table 1 has been deleted from the manuscript. Instead, we have incorporated some of the information from the initial Table 1 (country where study was conducted, study design, treatment status) into the subsequent tables. [Pages 20 to 28]

Please reorganize tables. For example for depression: group the data depending on assessment tool (ex: MINI, DSM, clinical evaluation, CESD....) (cut off can be added and info on validation might be too). Mention only the prevalence for 50+. Classify the publication by date publication.

Response:

We have reorganized the tables as suggested. E.g. For common mental disorders, we have grouped studies according to the measure used: PHQ-9, CES-D, MINI, geriatric depression scale, CIDI, and studies that used other methods. For each measure, we have also tried to list the studies according to the cut off used for quick reference and comparison. We note; however, this was especially challenging as the individual studies, e.g. for CES-D, used quite diverse cut-offs. For the prevalence estimates, we have only the specific details for PLWH50+ and their uninfected counterparts wherever this is applicable.

We have also classified the publications by date of publications as suggested in all the tables. However, in some cases the listing may be disordered as we had to also consider the measures used as was suggested by the first reviewer [Pages 20 to 28]

Authors need to reorganize the others tables in the same way.

Response:

We have also reorganized the other tables as suggested: incorporating some of the information from the initial table 1 and reordering the studies accordingly.

Is necessary to keep well-being in this paper? There is a lot of information. Maybe it could be presented in another paper? [EDITORS' NOTE: we are happy for the authors to rebut this request; perhaps the reporting could be made more concise instead?]

Response:

Thank you for the suggestion. We have chosen to maintain these aspects in the current paper. Although many of the studied outcomes could be considered individually, they are usually very interrelated. In the context of HIV and aging and the emerging nature of this subject in the region, presenting a comprehensive review offers better perspective regarding the research, clinical and policy implications as it indicates the need for people to take a more comprehensive approach in the treatment and management of OALWH. However, in the updated manuscript, we have endeavored to be more concise in our reporting.

In Obimakinde, age 60+ not 50+

Response:

We thank the reviewer for picking this out. We have corrected it accordingly in the revised manuscript.

How the authors compute the prevalence when they had middle-aged patients? It is not clear.

Response:

We have added a brief explanation in the paper's methods section under 'data extraction' for clarification. For studies with some middle-aged participants but whose mean/median age was ≥ 50 years, we extracted the reported percentages for the relevant outcomes for the whole sample. A good example here will be studies emanating from the UGANDAC cohort in Uganda and HAALSI cohort in South Africa, which included adults aged at least 40 years, but whose mean age was 52 and 61.7, respectively. For the studies that had middle-aged participants (but whose mean/median age was not at least 50 years), we computed percentages for participants who fell within the ≥ 50 category usually provided in the papers. In some studies, it was impossible to compute these percentages. Hence, the occurrence of a specific outcome was reported in the original effect measure, e.g. odds ratio, median or mean.

*The following excerpt has been added in the methods **"For studies exclusively carried out among PLWH50+, we computed or extracted the reported percentages of the relevant outcomes. Similarly, for studies with middle-aged participants but whose mean/median age was ≥ 50 years, we also computed or extracted the reported percentages for the relevant outcomes for the whole sample. For the studies with middle-aged participants (but whose mean/median age was not at least 50 years), we computed percentages for participants who fell within the ≥ 50 category, which was usually provided in the papers. In some studies, it was impossible to calculate these percentages. Hence, the occurrence of a specific outcome was reported in the original effect measure, e.g. odds ratio, median or mean."** [Fourth paragraph of methods section on page 4]*

5. Discussion/ conclusions

The conclusion repeats in part the first paragraph of the discussion.

Response:

We have rectified this error in the updated manuscript. The conclusion has been restructured succinctly to avoid repetition.

Discussion may be reorganized in parts: depression/anxiety, cognitive impairment, frailty to facilitate the reading.

Response:

We appreciate the reviewer for this helpful feedback. Indeed, we have restructured the discussion as follows: summary of main findings; depression/anxiety; cognitive impairment; frailty; quality of life; correlates of the mental and wellbeing outcomes, strength & limitations and conclusions. [Pages 10 to 14]

Reviewer 3: Dr. Makandwe Nyirenda, South African Medical Research Council

This is a very important paper in an area with limited empirical evidence. The paper clearly defined the study objectives and methodology. The study results and discussion are also clearly presented.

This paper makes an valuable contribution on the health of HIV-infected and uninfected older adults in sub-Saharan Africa.

My minor queries for the authors are:

1) In line 33 on page 6, you say 50 studies met the eligibility criteria and were include. However, in lines 34-39 the count of studies and their geographical location suggests more than 50 studies. Were some studies duplicated or multi-country studies? Please clarify.

Response:

We thank the reviewer for this observation. Indeed, a few of the studies were conducted in multiple countries hence the apparent mismatch in numbers. In the updated manuscript, we have provided this additional information. The following information has been captured in the results section: “However, some of these studies were conducted in multiple countries.” [Page 5 – 2nd last paragraph]

2) What effect, if any, would the inclusion in the search terms of the full name for HIV, i.e Human Immunodeficiency Virus, have made?

Response:

We acknowledge the reviewer's feedback. We tested this suggestion in a few of the databases e.g. PubMed and PsycINFO (with and without the suggested term) and there was very minimal differences in the number of hits received.

3) The authors need to acknowledge the limitation of excluding non-English language studies. Some crucial studies from Franco-phone sub-Saharan African countries may have been missed and hence results presented biased to that extent.

Response:

We have incorporated this suggestion in the updated manuscript accordingly. The following statement has been added in the limitations sections of the discussion: “Despite a thorough search strategy, some articles may have been missed especially in excluding non-English language studies. However, these were very few.” We appreciate the reviewer for this additional information.

VERSION 2 – REVIEW

REVIEWER	David van de Vijver Erasmus MC, Viroscience
REVIEW RETURNED	13-Aug-2021

GENERAL COMMENTS	Thank you for addressing my comments in a clear fashion.
--

REVIEWER	Charlotte Bernard University of Bordeaux
REVIEW RETURNED	24-Aug-2021

GENERAL COMMENTS	The authors have done a great job responding to comments and the manuscript is clearer.
---

	It is a very interesting and important work in the field. Three minor comments: - The authors should mention that Non-English studies are excluded in the search strategy - The title 3.2 should be Depression and anxiety In this section, the authors could mention the scales used for anxiety - In the discussion, the paragraph "Despite the observed limitations, the evidence of better mental health...." should be moved after paragraph 4.5
--	---

VERSION 2 – AUTHOR RESPONSE

Reviewer: 1

Dr. David van de Vijver, Erasmus MC

Comments to the Author:

Thank you for addressing my comments in a clear fashion.

Response:

We highly appreciate the reviewer for the constructive comments which have greatly improved our manuscript.

Reviewer: 2

Dr. Charlotte Bernard, University of Bordeaux Comments to the Author:

The authors have done a great job responding to comments and the manuscript is clearer. It is a very interesting and important work in the field.

Three minor comments:

-The authors should mention that Non-English studies are excluded in the search strategy

Response:

We appreciate the reviewer for the clarification. We have updated our manuscript accordingly. The following addition has been made in the search strategy: "We also excluded studies that were published in non-English languages." [Third paragraph of the methods section, page 4]

- The title 3.2 should be Depression and anxiety

Response:

We have adopted the suggestion and the current title is "Depression and anxiety" [Page 6] Mental health and Wellbeing of older adults living with HIV in sub-Saharan Africa: a systematic review

In this section, the authors could mention the scales used for anxiety

Response:

We appreciate the reviewer for this comment. We have added the relevant information in the said section. The following information has been added: "On the other hand, anxiety was assessed using the Schedule for Clinical Assessment in Psychiatry (SCAN) and from clinical records [30, 39]." [2nd last paragraph on page 6]

-In the discussion, the paragraph "Despite the observed limitations, the evidence of better mental health..." should be moved after paragraph 4.5

Response:

We have moved the text to the suggested position in the main text [last paragraph on page 12 on page 13].